# Azithromycin to prevent post-discharge morbidity and mortality in Kenyan children: a protocol for a randomised, double-blind, placebo-controlled trial (the Toto Bora trial)

Patricia B Pavlinac,[1] Benson O Singa,[2,3] Grace C John-Stewart,[1,4,5,6] Barbra A Richardson,[1,7] Rebecca L Brander,[4] Christine J McGrath,[1] Kirkby D Tickell,[1,3] Mary Amondi,[2] Doreen Rwigi,[2] Joseph B Babigumira,[1,8] Sam Kariuki,[9] Ruth Nduati,[10] Judd L Walson[1,3,4,5,6]

For numbered affiliations see end of article.

**Correspondence to**
Dr Patricia B Pavlinac;
ppav@uw.edu

## ABSTRACT

**Introduction** Child mortality due to infectious diseases remains unacceptably high in much of sub-Saharan Africa. Children who are hospitalised represent an accessible population at particularly high risk of death, both during and following hospitalisation. Hospital discharge may be a critical time point at which targeted use of antibiotics could reduce morbidity and mortality in high-risk children.

**Methods and analysis** In this randomised, double-blind, placebo-controlled trial (Toto Bora Trial), 1400 children aged 1–59 months discharged from hospitals in Western Kenya, in Kisii and Homa Bay, will be randomised to either a 5-day course of azithromycin or placebo to determine whether a short course of azithromycin reduces rates of rehospitalisation and/or death in the subsequent 6-month period. The primary analysis will be modified intention-to-treat and will compare the rates of rehospitalisation or death in children treated with azithromycin or placebo using Cox proportional hazard regression. The trial will also evaluate the effect of a short course of azithromycin on enteric and nasopharyngeal infections and cause-specific morbidities. We will also identify risk factors for postdischarge morbidity and mortality and subpopulations most likely to benefit from postdischarge antibiotic use. Antibiotic resistance in *Escherichia coli* and *Streptococcus pneumoniae* among enrolled children and their primary caregivers will also be assessed, and cost-effectiveness analyses will be performed to inform policy decisions.

**Ethics and dissemination** Study procedures were reviewed and approved by the institutional review boards of the Kenya Medical Research Institute, the University of Washington and the Kenyan Pharmacy and Poisons Board. The study is being externally monitored, and a data safety and monitoring committee has been assembled to monitor patient safety and to evaluate the efficacy of the intervention. The results of this trial will be published in peer-reviewed scientific journals and presented at relevant academic conferences and to key stakeholders.

**Trial registration number** NCT02414399.

### Strengths and limitations of this study

► Randomised, placebo-controlled, double-blinded design and modified intention-to-treat analysis will ensure unbiased treatment effect measure.
► Comprehensive data are collected, including biological specimens for all child participants and a subset of adult caregivers, for analyses of mechanisms of postdischarge morbidity and mortality, subsets of children most likely to benefit from the antibiotic, as well as assessments of antibiotic resistance and cost-effectiveness.
► Results will likely be generalisable due to the limited exclusion criteria, large sample size and multiple study sites
► The primary endpoint of this study is a combined outcome of rehospitalisation and death which, while improving statistical power, may present challenges for interpretation.
► Children in both intervention arms may receive other antibiotics over the course of follow-up.

## BACKGROUND

Close to 3 million deaths occur annually in children less than 5 years of age in sub-Saharan Africa (SSA), over half of which are attributed to infectious causes.[1] Children who were recently hospitalised have mortality rates sixfold to eightfold higher than similarly aged children from the same community.[2–4] Post-discharge mortality rates as high as 15% have been documented in the 12 months following discharge, with mortality risk remaining elevated up to 2 years postdischarge.[5–9] Children who are very young, malnourished or HIV infected are at particularly high risk of postdischarge mortality within the 3 months following discharge.[2–5 7–9] Children being

discharged from hospital in SSA may represent an accessible high-risk population in which to target interventions to reduce mortality and morbidity.

Targeted antibiotic interventions, including the use of cotrimoxazole among HIV-infected children and the use of amoxicillin or cefdinir among children with severe acute malnutrition (SAM), have been shown to reduce morbidity and mortality in these specific vulnerable populations.[10–13] Other trials of targeted antibiotic use in vulnerable populations, including cotrimoxazole in HIV-exposed uninfected children and in children with SAM, have failed to demonstrate a mortality benefit.[14 15] In contrast, non-targeted mass drug administration of a single dose of azithromycin halved mortality rates among Ethiopian children living in communities randomised to receive the antibiotic.[16 17] Concerns about the potential emergence of antibiotic resistance, possible toxicity and feasibility of delivery are barriers to community-wide antibiotic distribution strategies.

A short course of azithromycin given to children with recent severe illness being discharged from hospital may optimise benefit while reducing both individual and population level risks. Azithromycin may reduce postdischarge morbidity and mortality through infection-related mechanisms such as treating undiagnosed, incompletely treated or nosocomial infections, or by protecting against new or recrudescent infections that occur during recovery. Azithromycin may also act through non-antimicrobial pathways by anti-inflammatory and/or immune-modulatory effects.

## OBJECTIVE

The primary objective of this double-blind, placebo-controlled, randomised controlled trial (RCT) is to determine whether a 5-day course of azithromycin in children age 1–59 months discharged from hospital in Western Kenya reduces rates of rehospitalisations and/or death in the subsequent 6 months. The secondary objectives are (1) to evaluate possible mechanism(s) by which azithromycin may affect morbidity and mortality by comparing reasons for rehospitalisation and prevalence of enteric and nasopharyngeal infections between the randomisation arms; (2) to determine whether empirical administration of azithromycin at hospital discharge increases risk of antibiotic resistance in commensal *Escherichia coli* and *Streptococcus pneumoniae* isolates from treated children and their primary caregivers; (3) to identify correlates and intermediate markers of postdischarge mortality and hospital readmission; (4) to determine the cost-effectiveness of postdischarge administration of a 5-day course of azithromycin in settings of varying antibiotic use, rehospitalisation rates and mortality rates; and (5) to create a repository of stool, nasopharyngeal and blood specimens from highly characterised, recently discharged children, half of whom are treated with azithromycin to be used to address future research questions.

## METHODS

Reporting of this study protocol has been verified in accordance with the Standard Protocol Items for Randomised Trials recommendations.

### Eligibility

Children aged 1–59 months old weighing at least 2 kg who have been hospitalised, and subsequently discharged, will be eligible for inclusion. Caregivers of potentially eligible children must be at least 18 years of age or classified as an emancipated minor and be willing to participate in the contact cohort if randomly selected. Children will be excluded if: azithromycin is contraindicated (children taking or prescribed other macrolide antibiotics, such as erythromycin or clarithromycin, or the protease inhibitor lopinavir); they were admitted to hospital for a trauma, injury or a birth defect; they do not plan to remain in the study site catchment area for at least 6 months; the legal guardian does not provide consent; or if a sibling was enrolled in the trial on the same day of discharge.

### Recruitment

Children will be recruited from the inpatient wards of health facilities in Kisii and Homa Bay Counties where study staff will accompany hospital staff on ward rounds to identify children being discharged each day. All discharged children, as determined by the onsite hospital clinicians, will be screened by study staff during working hours. If the caregiver is interested in participating and indicates consent for screening, the study staff will screen the child for eligibility, and if eligible, will obtain informed consent for study participation. Informed consent includes an explanation of the potential risks and benefits of the study and additional provision for use of participant data and samples for future studies, and will be conducted in the language of the respondent's choosing (English, Kiswahili, Kisii or Luo). The parent or guardian (primary caregiver) must sign written informed consent (or provide a witnessed thumbprint if not literate) prior to enrolment.

### Enrolment

Children will be enrolled at the time of discharge by the clinical staff. At enrolment, primary caregivers will be interviewed to assess demographic information, medical history and detailed contact information for the child (table 1). Information about the hospitalisation will also be abstracted from medical records, including presenting diagnosis, medical management, length of stay, procedures performed, relevant medical history, physical examination and laboratory data. All enrolled participants will undergo a physical examination performed by the study clinician, including measurement of height (in children ≥24 months), length (in children <24 months), weight and midupper arm circumference (MUAC), each of which will be measured twice. HIV status will be obtained from medical records or from performed testing if records are not available. Detailed

**Table 1** Summary of data collected among enrolled children at each study visit

| Enrolment visit (hospital discharge) | 3-month follow-up visit | 6-month follow-up visit | Unscheduled visits |
|---|---|---|---|
| ► Questionnaire of sociodemographic, clinical history, treatments prescribed in hospital and at discharge, hospitalisation costs, dietary factors, household factors and environmental exposures.<br>► Physical exam.<br>► Anthropometry.<br>► Abstraction of medical records if available.<br>► Heel/finger prick (HIV and malaria).<br>► Stool collection (*Shigella*, *Salmonella*, *Campylobacter*, *Escherichia coli*, *Cryptosporidium* and *Giardia*).<br>► Nasopharyngeal swab collection (*Streptococcus pneumoniae*). | ► Questionnaire of study drug administration and reported illnesses, hospitalisation costs if rehospitalised, change in clinical history and treatments since last visit.<br>► Physical exam.<br>► Anthropometry.<br>► Abstraction of medical records if available (if rehospitalised).<br>► Verbal autopsy (or abstracted medical records).<br>► Heel/finger prick (HIV and malaria).<br>► Stool collection (*Shigella*, *Salmonella*, *Campylobacter*, *E. coli*, *Cryptosporidium* and *Giardia*).<br>► Nasopharyngeal swab collection (*S. pneumoniae*). | ► Questionnaire of reported illnesses, hospitalisation costs if rehospitalised, change in clinical history and treatments since last visit.<br>► Physical exam.<br>► Anthropometry.<br>► Abstraction of medical records if available (if rehospitalised).<br>► Verbal autopsy (or abstracted medical records).<br>► Heel/finger prick (HIV and malaria, sickle cell).<br>► Stool collection (*Shigella*, *Salmonella*, *Campylobacter*, *E. coli*, *Cryptosporidium* and *Giardia*).<br>► Nasopharyngeal swab collection (*S. pneumoniae*). | ► Questionnaire of reported illnesses since last scheduled visit, change in clinical history and treatments since last visit.<br>► Abstraction of medical records if available (if rehospitalised).<br>► Verbal autopsy (or abstracted medical records). |

home location and contact information will be collected to enable patient tracing.

## Specimen collection

Specimens will be collected at enrolment (prior to study medication administration, as well as at 3-month and 6-month follow-up visits). All children will also be asked to provide a whole stool for enteric pathogen identification and storage. Stool samples will be divided within 1 hour of collection for the following purposes: (1) placed in Cary-Blair for eventual bacterial culture (FecalSwab Cary-Blair Collection and Transport Systemt, Copan Diagnostics, Murrieta, California, USA), (2) immediately tested for *Giardia* and *Cryptosporidium* using the immunoassay (Quik Chek, Alere, Waltham, Massachusettes, USA) and (3) placed in –80°C storage for future molecular determination of pathogen or commensal flora and markers of gut function (whole stool for these purposes will be divided into two separate vials). If a child cannot produce whole stool, two flocked rectal swabs (Pedatric FLOQswab, Copan Diagnostics) will be collected: one placed in Cary-Blair and the other stored in –80°C for future analyses.

One flocked dry nasopharyngeal swab (Copan Diagnostics) will also be collected from all enrolled children at each time point, immediately placed in skim milk, tryptone, glucose and glycerine media and frozen (–80°C) within 1 hour of collection for future *S. pneumoniae* culture.[18 19] Primary caregivers in the contact cohort will also be asked to provide a stool sample (or 2 rectal swabs) and nasopharyngeal sample at each visit for testing and storage as described above.

Venous blood (up to 1 teaspoon (5 mL)) will be collected from all enrolled children at each time point into EDTA tubes and separated for the following purposes: (1) 0.5 mL for immediate HIV testing (if indicated according to Kenyan Ministry of Health guidelines), (2) 0.4 mL for a thin malaria smear, which will be stored at room temperature, (3) 0.4 mL for a dried blood spot and (4) 2–4 mL for plasma and buffy coat isolation and –80°C storage. Blood will also be collected from primary caregivers for HIV testing if indicated.

## Randomisation

Block randomisation (1:1) in random-sized blocks of no more than 10, stratified by site, will be used. Primary randomisation will include allocation to the contact cohorts at a ratio of 1:5 (resulting in 150 per treatment arm). Each subject will be assigned a patient identification (PID) number, and the randomisation code linking each PID to the allocated treatment will be generated by a designated statistician and maintained by the University of Washington Research Pharmacy. Study participants, investigators (other than the statistician), the study staff, hospital clinicians and persons involved in data management or analysis will remain blinded to the allocation group during all data collection phases of the study.

| Weight (kg) | Day 1 dose (mL) | Day 2–5 dose (mL) |
|---|---|---|
| 2.0 | 0.25×2 | 0.25 |
| 2.1–2.4 | 0.30×2 | 0.30 |
| 2.5–2.8 | 0.35×2 | 0.35 |
| 2.9–3.2 | 0.40×2 | 0.40 |
| 3.3–3.6 | 0.45×2 | 0.45 |
| 3.7–4.0 | 0.50×2 | 0.50 |
| 4.1–4.8 | 0.60×2 | 0.6 |
| 4.9–5.6 | 0.70×2 | 0.7 |
| 5.7–6.8 | 0.85×2 | 0.85 |
| 6.9–8.0 | 1.0×2 | 1.0 |
| 8.1–9.6 | 1.2×2 | 1.2 |
| 9.7–11.2 | 1.4×2 | 1.4 |
| 11.3–13.6 | 1.6×2 | 1.6 |
| 13.7–16.0 | 2.0×2 | 2.0 |
| 16.1–19.2 | 2.4×2 | 2.4 |
| 19.3–23.2 | 2.9×2 | 2.9 |
| 23.3–25.0 | 3.2×2 | 3.2 |

Table 2  Azithromycin dosing chart by child weight

## Intervention

Enrolled children will be prescribed a 5-day course of oral suspension formulation azithromycin (Zithromax from Pfizer, 10 mg/kg on day 1, followed by 5 mg/kg/day on days 2–5) or identically appearing and tasting placebo at discharge. Identically appearing bottles will be prelabelled with the PID. Dosing ranges were determined such that a given child would never be underdosed or overdosed by more than 20% of the weight-specific intended dose (table 2). The day 1 dose will be split in half, and the first half will be administered by the study clinician (to be observed by the caregiver), followed by the second half administered by the caregiver under careful observation of the study staff. Day 2–5 doses will be administered by caregivers at their home. Caregivers will be provided with an instruction sheet including visual and written instructions in the language of their choosing (English, Kiswahili, Luo or Kisii).

Automated daily text message drug administration reminders will be sent for the 4 days following discharge, and caregivers will be asked to respond with whether the child took the daily dose. The response text message will be free of charge to caregivers, and caregivers will be reimbursed for each response at the final study visit. Caregivers are also asked to record each administered dose on the bottle and to return bottles at the 3-month follow-up visit. The questionnaire administered during the 3-month follow-up visit also includes questions about how many doses of the study drug the child received.

## Follow-up procedures

All enrolled children and primary caregivers will be scheduled to return to the health facility at 3 and 6 months following enrolment to collect clinical information and samples. Anthropometric measurements will be obtained from all children and caregivers at both follow-up visits (height/length, weight and MUAC), and caregivers will be asked about any hospitalisations occurring since the last time the child was seen by study staff. Caregivers will be provided with 400KSH (approximately US$4) to cover the cost of their round-trip transportation.

If the participant does not return at their scheduled time, study staff will attempt to make contact with the primary caregiver via cell phone; if no telephone number is provided, or if the participant cannot be reached, study staff will trace the child to the household within 2 weeks of the scheduled follow-up time.

During scheduled follow-up visits, study staff will use a standardised questionnaire to ascertain history of recent illness/morbidity, postdischarge medication use including antibiotic treatment and current condition of the child (any hospitalisations, admission and discharge date of any hospitalisation, vital status and date of death if applicable). If caregivers report a hospitalisation, causes of admission, medication administration, cost of admission and length of stay will be ascertained from both caregivers and medical records, when available.

Caregivers will be encouraged, at enrolment and at each subsequent contact, to bring the child to the study health facility at any time the child is sick. Study staff will triage children to the appropriate health facility staff and will conduct a brief unscheduled visit questionnaire to ascertain adverse event information. If the unscheduled visit leads to a hospitalisation, this will trigger the completion of a hospital admission form (abstraction of medical records).

If at any point during follow-up a child dies, a verbal autopsy using the Population Health Metrics Research Consortium Shortened Verbal Autopsy Questionnaire[20] will be administered. If the death occurred in a hospital, data from the hospital records, including cause of death, if available, will be abstracted. If a death certificate is available, cause(s) and timing of death will be abstracted.

Final causes of rehospitalisation and death will be determined after data collection is complete by an independent adjudication committee comprised of clinicians specialising in paediatrics and infectious disease. Sources of cause of rehospitalisation (medical records and caregiver report) and causes of death (causes automatically assigned from the verbal autopsy using SmartVA-Analyze [Tariff 2.0 Method],[21] hospital records or death certificates) will be presented to the adjudication committee for final cause assignment.

## Laboratory procedures

Stool/rectal swabs, nasopharyngeal swabs and blood will be collected as described above and undergo either immediate or future laboratory testing as described in table 3. All biological samples will be collected by staff trained in biosafety and Good Clinical Laboratory Practice. Samples will be processed in Kenya when technology is available

**Table 3** Sample storage and processing descriptions

| Specimen collected | Purpose | Tests performed |
|---|---|---|
| Stool/flocked rectal swabs | Bacterial ID and storage for AST | Fresh samples/rectal swabs will be cultured to identify *Shigella* spp., *Salmonella* spp., *Campylobacter* spp. and *Escherichia coli* using standard microbiological methods and biochemically confirmed using bioMérieux's API strips. All *Shigella* spp., *Salmonella* spp. and *Campylobacter* spp. isolates, as well as a random subset of *E. coli* isolates will undergo antibiotic resistance testing using disc diffusion for the following antibiotics: amoxicillin–clavulanic acid (augmentin), ampicillin, azithromycin, chloramphenicol, ciprofloxacin, ceftriaxone, ceftazidime, cefoxitin, gentamicin, imipenem, trimethoprim-sulfamethoxazole, ceftazidime/clavulanate (ESBL), cefotaxime/clavulanate (ESBL). Categorisations of susceptible, intermediate and resistant will be determined using zone-size cut-offs outlined in CLSI interpretive standards. |
| | Parasite detection | Fresh stool and rectal swabs will be tested for *Giardia* and *Cryptosporidium* using the immunoassay *Giardia/Cryptosporidium* QUIK CHEK. |
| | Storage | Stool/flocked swabs and colonies of *E. coli*, *Shigella* spp., *Salmonella* spp. and *Campylobacter* spp. will be stored at −80°C. |
| Nasopharyngeal swabs | Bacterial isolation, storage and resistance testing | *S. pneumoniae* colonies will be isolated using standard microbiological or molecular diagnostic protocols and susceptibility testing performed using standard microbiological or molecular techniques. A random subset of *S. pneumoniae* isolates will undergo antimicrobial resistance testing using disc diffusion for the following antibiotics: amoxicillin–clavulanic acid (Augmentin), ampicillin, azithromycin, chloramphenicol, ciprofloxacin, ceftriaxone, ceftazidime, cefoxitin, imipenem and trimethoprim–sulfamethoxazole. Categorisations of susceptible, intermediate and resistant will be determined using zone-size cut-offs outlined in CLSI interpretive standards. |
| | Storage | Back-up sample and *S. pneumoniae* colonies will be will be stored at −80°C. |
| Blood | HIV and malaria testing | HIV testing will be performed per Kenyan National Guidelines and malaria microscopy performed using standard methods. |
| | Storage | Plasma and buffy coat will be stored at −80°C. Dried blood spots will be stored at room temperature. |

AST, antibiotic susceptibility testing; CLSI, Clinical and Laboratory Standards Institute; *E.coli, Escherichia coli;* ID, identification; *S. pnuemoniae, Streptococcus pneumoniae;* spp., species.

at the Kenya Medical Research Institute (KEMRI) Wellcome Trust or Centre for Microbiology Research. Metagenomic analyses and/or analyses that require technology not available in Kenya will be performed at the University of Washington. If stool culture results report *Shigella* or *Salmonella* infection, the study staff will contact the child's caregiver and encourage the caregiver to bring the child back for an evaluation and potential treatment if the child is symptomatic.

### Data management and confidentiality
Personal information about the participants, including medical records and data ascertained per caregiver interview, will be securely stored in files in the study offices at the study sites. Only predesignated study staff will have access to the files. Data will be entered into an electronic database (Dacima Electronic Data Capture) regularly by study staff. Access to the electronic database will be secured using password-protected accounts for study staff. Data reports of screening, enrolment, exclusion and follow-up totals will be disseminated to the study team on a weekly basis; reports including baseline demographic characteristics, laboratory results, adherence data and serious adverse event (SAE) summaries will be distributed to study coinvestigators and data monitors quarterly. Data will be regularly queried to facilitate ongoing data cleaning.

### Data analysis
#### Primary endpoints
The primary study endpoint is a combined outcome of mortality and hospital readmission, as rehospitalisation is highly associated with risk of subsequent poor outcome.[2] Rehospitalisations that are a continuation of management from the previous hospitalisation (such as elective blood transfusion) or that occur during enrolment

procedures due to a clinical deterioration postdischarge, will be excluded from the analysis. Loss to follow-up will be defined as non-attendance at both follow-up visits despite 1 month of active tracing and no clear evidence of death.

## Secondary endpoints

1. Cause-specific rehospitalisations assessed by questionnaire (maternal recall of diagnosis) at month 3 and month 6 follow-up visits and by medical record review (discharge diagnosis). In cases when both sources are available, information from the medical record will be considered as the primary source. Separate analyses will be performed for each diagnosis: diarrhoea, acute respiratory infection, malnutrition or malaria.

2. Mild, moderate and severe events that did not result in rehospitalisation, including diarrhoea, vomiting, skin rash, lip swelling, difficulty breathing/wheeze and seizure will be ascertained by caregiver report or identified by the study clinicians during clinical exams at scheduled follow-up visits or during unscheduled visits. Severity (grades 1–3) will be defined according to 2014 Division of AIDS Table for Grading the Severity of Adult and Pediatric Adverse Events.

3. Enteric pathogen carriage, operationalised as presence of a bacterial pathogen—*Shigella* species (spp.), *Campylobacter* spp. or *Salmonella* spp.—or parasite—*Giardia* or *Cryptosporidium*—in stool or rectal swabs assessed at month 3 and month 6 follow-up visits.

4. *S. pneumoniae* isolated from nasopharyngeal swab cultures at month 3 and month 6 follow-up visits.

5. Antibiotic resistance, specifically resistance to azithromycin, ampicillin, augmentin, ciprofloxacin, trimethoprim–sulfamethoxazole, in *E. coli* and *S. pneumoniae* isolates, and presence of extended spectrum beta-lactamase (ESBL) in *E. coli* isolates, from month 3 and month 6 follow-up visits.

## Statistical analysis

### To compare rates of rehospitalisation and mortality in the 6 months following hospital discharge among Kenyan children receiving 5-day azithromycin versus placebo

Primary analyses will be modified intent-to-treat (mITT) based on randomisation allocation to the 5-day course of azithromycin versus placebo. Cumulative incidence of death or first rehospitalisation will be compared between treatment groups using Cox proportional hazards regression. Participants will be censored at the date of their first rehospitalisation or at the date of death. Median time to hospitalisation-free survival will be compared between randomisation groups using Kaplan-Meier (K-M) survival analysis and associated log-rank test. If the baseline assessment of randomisation reveals an imbalance in characteristics between the treatment groups, we will evaluate these variables as potential confounders in a subanalysis secondary to the mITT. Potential baseline confounders will be added stepwise in a multivariable Cox model and

maintained in the model if adjustment changes the HR by more than 10%. In per-protocol analyses also secondary to the mITT, we will compare treatment effects in groups defined by self-reported adherence to the 5-day course of azithromycin (5 doses vs <5 doses; ≥3 doses vs <3 doses; >1 dose vs 1 dose only). In addition, we will conduct Cox regression and K-M survival analyses for time to mortality and time to rehospitalisation as separate endpoints to understand intervention effects on these outcomes individually. The assumption of proportional hazards will be checked in all models using graphical methods including plotting an $\ln(-\ln(S(t)))$ plot for each treatment group and assessing the parallelism of the 2 lines and by plotting Schoenfeld residuals over time. If there is substantial missing covariate data, multiple imputation using the Markov chain Monte Carlo method will be used to impute covariate information. Missing outcome data (death or rehospitalisation) will not be imputed, but participants will be censored at the last follow-up visit therefore contributing some person-time to the analysis. In sensitivity analyses, we will compare treatment effects in children whose caregivers report no additional antibiotic use over follow-up and separately, who report no additional azithromycin use specifically and in subsets of children defined by age, site and discharge diagnosis.

### To evaluate possible mechanism(s) by which azithromycin may affect morbidity and mortality by comparing reasons for rehospitalisation and change in prevalence of pathogen carriage between the randomisation arms

To evaluate the association between azithromycin and the rates of cause-specific rehospitalisations (hospitalisation due to diarrhoea, acute respiratory infection, malnutrition or malaria), we will use Anderson-Gill proportional hazards modelling with previous rehospitalisations included as time-dependent covariates in the model to capture the dependent structure of recurrence times. Because we will not have granularity in the time points other than 3 months and 6 months for assessment of pathogen carriage, we will compare the prevalence of a bacterial and parasitic pathogens (*Shigella* spp., *Salmonella* spp., *Campylobacter* spp., *Cryptosporidium* spp. and *Giardia*) at 3 and 6 months by randomisation arm using generalised estimating equations (GEE) with a Poisson link, exchangeable correlation structure, and will adjust for baseline presence of a bacterial pathogen. To determine whether an observed association between the intervention and pathogen carriage wanes over time, we will test the hypothesis that the prevalence ratios comparing carriage in intervention arms are the same at the 2 follow-up time points using a $\chi^2$ test.

### To determine whether empirical administration of azithromycin at hospital discharge increases risk of antibiotic resistance in commensal E. coli and pneumococcal isolates from treated children and their household contacts

Among children and adult household contacts in whom commensal *E. coli* and/or *S. pneumoniae* are isolated,

we will compare the proportion of isolates resistant to azithromycin, ampicillin, augmentin, ciprofloxacin and trimethoprim–sulfamethoxazole, between randomisation arms and contact cohorts for each arm, at 3 and 6 months using GEE with a Poisson link and exchangeable correlation structure. A $\chi^2$ test will be used to determine whether the association between intervention arm and antibiotic resistance wanes over time. Because the likelihood of having a bacterial pathogen isolated may depend on baseline factors, including intervention arm, we will conduct secondary analyses using propensity scores to account for the potential differential likelihood of having antibiotic susceptibility testing performed, which will allow us to make inference to the entire study population and their contacts. Also, we will compare resistance proportions among children (as opposed to among isolates) where absence of an isolated bacteria is considered not resistant.

### To identify correlates and intermediate markers of postdischarge mortality and hospital readmission among hospitalised children

Enrolment hospital admission diagnosis, indicators of malnutrition, age, HIV-exposure and HIV-infection status, sickle cell anaemia and randomisation arm will be assessed in a multivariable Cox regression model to identify correlates of the primary endpoint of death and/or hospital readmission independent of the treatment effect. In addition, Cox regression models will also be built for correlates of mortality and correlates of readmission separately to understand distinct cofactors for each of these outcomes.

### To determine the cost-effectiveness of postdischarge administration of a 5-day course of azithromycin in settings of varying antibiotic use, rehospitalisation rates and mortality rate

Costs analysis: we will assess the costs of all supplies, services and equipment necessary to implement the intervention (direct medical costs). The perspective will be that of the healthcare provider, that is, Kenya's Ministry of Health. Using WHO guidelines and its ingredients approach, we will quantify the resources and associated unit costs required to deliver a 5-day course of azithromycin, organised in standard expenditure categories: personnel (salaries), supplies, including drugs, equipment, services, space and overhead. We will also measure the costs of severe child hospitalisations, the costs for the different types of personnel employed (eg, nurses/doctors) and the time demanded from them for conducting the intervention.[22] When data are missing, they will be complemented by data extracted from the literature and other available sources. Full incremental costs will be derived, with estimation of the potential healthcare cost-offset realised in avoiding severe hospitalisations. Costs will be measured in local currency (Kenyan Shilling) and converted into US\$. Our main metric will be cost per child treated. Cost-effectiveness analysis (CEA): we will develop a CEA mathematical model and estimate

incremental costs and cost-effectiveness for implementation of the intervention. The model will include two components: costs (described immediately above) and health benefits. The study will provide clinical outcomes (mortality/morbidity) over a 6-month follow-up period. Subsequently, deaths averted, life-years saved and disability-adjusted life years (DALYs) averted by the intervention will be estimated. We will estimate: (A) incremental costs and (B) incremental cost-effectiveness of the intervention versus status quo. *Incremental costs* are the net sum of the costs to implement the intervention compared with status quo and the costs averted due to the decrease in severe child hospitalisations. *Incremental cost-effectiveness* ratios will be estimated as cost per death averted, cost per life-year saved and cost per DALY averted. We will use recent estimates of disability weights for estimation of DALYs.[23 24] Short-term (over study follow-up, ie, 6 months) and longer term time horizons (extrapolated to 1, 5 and 10 years) will be used. DALYs and costs will be discounted at 3% per year, consistent with CEA guidelines (undiscounted results will also be presented). Sources of uncertainty in the results will be explored in univariate and probabilistic sensitivity analysis.[25 26] Finally, we will compare our findings to CEA estimates for other health interventions in SSA.[27 28]

### Data and safety monitoring

A data safety and monitoring committee (DSMC) will be established at study initiation to monitor SAEs and to evaluate the statistical analysis plan and associated stopping rules. The DSMC will include expertise in clinical trials, statistics, child mortality assessment, ethics and paediatric care in resource limited settings. Adverse events will be monitored by the DSMC. Monthly adverse event summaries will be sent to the DSMC safety officer, and individual child SAE forms, which include detailed medication history to evaluate possible drug interactions, will be sent to the safety officer per request. Each SAE will be assigned the plausibility of relatedness to study drug by study Principal Investigators (PIs). The data will not be presented by intervention group unless requested by DSMC safety officer. These reports will be descriptive (no statistical analyses). The DSMC will make recommendations regarding any imbalances in safety outcomes.

A single interim analysis for rehospitalisation-free survival will be prepared by the study statistician using O'Brien-Fleming boundaries for benefit and harm when 50% of expected person time (350 child-years) has been accrued. Assuming 157 events will be available at half of the person-time accrual, a z-score critical value of 2.797, or P value <0.005, from a K-M log-rank test will determine the cut-off of statistical significance. A symmetric boundary will be used for benefit and harm. The DSMC will review this analysis and make a determination about study continuation. Futility will not be a basis for stopping rules because of the trials' value in understanding

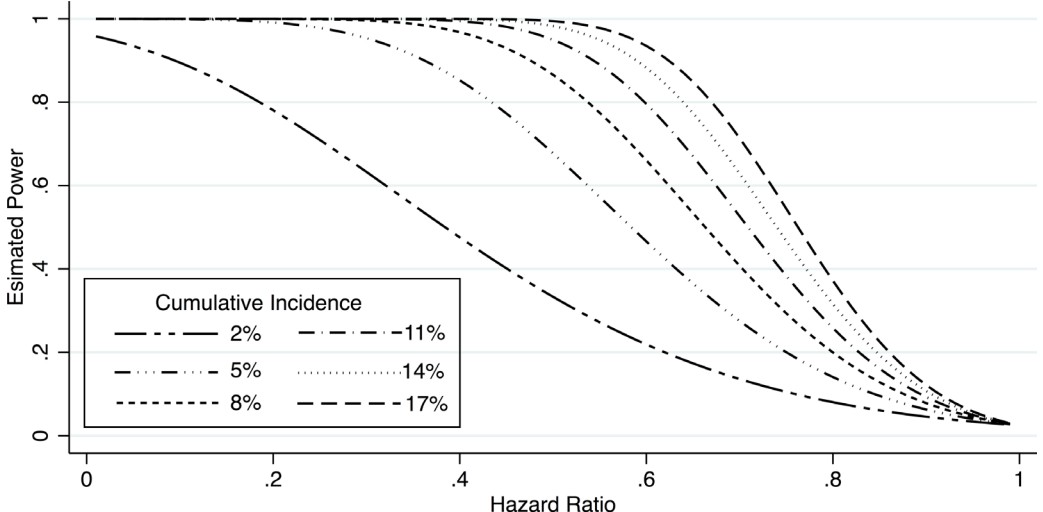

**Figure 1** Power and detectable HRs assuming a range of mortality rates from 2% to 17%.

mechanisms of postdischarge worsening and antibiotic resistance. Assuming the DSMC decides to continue the trial after the interim analysis, an alpha of 0.045 will be used as the statistical significance boundary at the final analysis.

### Statistical power

#### To compare rates of rehospitalisation and mortality in the 6 months following hospital discharge among Kenyan children receiving 5-day azithromycin versus placebo

The total sample size required was calculated for the primary endpoint of time to death or hospital readmission within the 6-month postdischarge period, assuming an alpha level of 0.05, power of 0.80 and a ratio of treatment to placebo random assignment of 1:1. In SSA, it is estimated that 2%–15% of children aged less than 5 years died within 6 months of hospital discharge and 15.5% of children who survived discharge from the district hospital were readmitted with the same diagnosis within 6 months.[2 4 9] Assuming that an additional 5%–10% of children are readmitted for other conditions, we expect that rehospitalisations will occur in 20.5%–30.5% of children enrolled in the study. Combined with our expected fatality rate (2%–15%), we expect the cumulative incidence of the combined endpoint to range from 22.5% to 45.5%.[9] Based on a previous trial of mass drug administration of a single dose of azithromycin in which a single dose of the antibiotic was associated with a 49% reduction in risk of death, we calculated sample sizes using estimates of reduction in risk ranging from 30% to 50% with the cumulative incidence of 22.5%–45.5% in the placebo-treated group and found the sample size required ranged from 90 to 550 children per treatment arm.[16] Using the most conservative estimates of a hazard ratio (HR) of 0.70 and 22.5% prevalence of readmission/death, we need to enrol 1100 children in the study (550 per arm) to achieve adequate power. We will recruit an additional 300 children (≈20%) to account for possible loss to follow-up, resulting in a total planned enrolment of 1400 children or an expected 700 per treatment group. When considering mortality alone, and estimated mortality ranges of 2%–17% among place-treated children, we will have >80% power to detect HRs ≤0.5 for mortality rates of ≥8% and HRs ≤0.6 for mortality rates ≥11% (figure 1).

#### To evaluate possible mechanism(s) by which azithromycin may affect morbidity and mortality by comparing reasons for rehospitalisation and prevalence of pathogen carriage between the randomisation arms

We calculated the minimum detectable association between treatment arm and cause-specific rehospitalisations (hospitalisation due to diarrhoea, acute respiratory infection, malnutrition or malaria vs any other) among enrolled children, assuming an alpha level of 0.05, power of 0.80 and a ratio of treatment to placebo of 1:1. Based on data from Kenya, rehospitalisation rates due to specific causes ranged from approximately 0.5%–5.7% in the 6-month postdischarge period.[4] By not conditioning on the child having the same diagnosis as the initial hospitalisation, we expect the cumulative incidence of cause-specific rehospitalisations to range from 2.5% to 10%. With this range of outcome rates, we will be able to detect HRs of 0.48–0.70 for the effect of azithromycin on specific severe morbidities.

We expect 56% of children in the placebo group to have *S. pneumoniae* isolated from nasopharyngeal swabs, providing ≥80% power to detect a prevalence ratio of 0.85 (or 1.15) between the two treatment arms at each time point.[29–31] Based on prevalences of *Shigella* spp., *Salmonella* spp., *Campylobacter* spp., *Cryptosporidium* spp. and *Giardia* among asymptomatic children in Western Kenya, we expect 10% of children in the placebo group to have a bacterial pathogen isolated at each time point, resulting in ≥80% power to detect ratios

**Table 4** Power (%) to detect prevalence ratios of macrolide and β-lactamase resistance in 200 *Escherichia coli* and 200 *Streptococcus pneumoniae* isolates per treatment group

| | | Resistance prevalence (%) in placebo group | | | | | | |
|---|---|---|---|---|---|---|---|---|
| Resistance prevalence (%) in azithromycin group | | 10 | 20 | 30 | 40 | 50 | 60 | 70 |
| | 10 | | | | | | | |
| | 20 | 80 | | | | | | |
| | 30 | >99 | 64 | | | | | |
| | 40 | >99 | 99 | 55 | | | | |
| | 50 | >99 | >99 | 98 | 48 | | | |
| | 60 | >99 | >99 | >99 | >99 | 52 | | |
| | 70 | >99 | >99 | >99 | >99 | 98 | 55 | |
| | 80 | >99 | >99 | >99 | >99 | >99 | 99 | 64 |

in enteric pathogen prevalences of 0.67 (1.49) at each time point.[32]

### To determine whether empirical administration of azithromycin at hospital discharge increases risk of antibiotic resistance in commensal *E. coli* and *S. pneumoniae* isolates from treated children and their household contacts

We will select a random selection of 400 *E. coli* and 400 *S. pneumoniae* isolates (200 per arm) for β-lactam and macrolide resistance testing at each time point. We will also store all *S. pneumoniae* and *E. coli* isolates and other isolated bacteria from stool for potential future testing in the event that resistance prevalence is lower than expected. As shown in table 4, we will have >80% power to detect prevalence ratios >1.1, with an ability to detect the smallest effect sizes when the prevalence of resistance in the placebo group is highest. We will enrol 300 adults in the contact cohort for *E. coli* and *S. pneumoniae* isolation. We expect *E. coli* to be isolated from all adults and *S. pneumoniae* isolated from between 5% and 55%.[29 33 34] Assuming an alpha of 0.05, a 1:1 ratio of testable isolates and a prevalence of resistance of 50% in the placebo arm, we will have 80% power to detect a 1.4-fold to 1.9-fold higher resistance prevalence in the contacts of azithromycin-treated children.

### To identify correlates and intermediate markers of postdischarge mortality and hospital readmission among hospitalised Kenyan children

Conservatively estimating a 20% loss-to-follow-up rate in the RCT and a cumulative incidence of death or rehahospitalisation of 22.5%, we will have >80% power to detect HRs ≥1.3 between correlates and the outcome with exposure prevalences of ≥20% or more and HRs ≥1.5 for exposure prevalences <20%.

### Study timeline

The trial began on 28 June 2016, and participant recruitment and follow-up will continue over a 36-month period, with anticipated final follow-up visit(s) occurring in June 2019. Primary analyses will be complete by February 2020.

### Potential challenges and limitations

In order to ensure adequate power to detect a discernible clinically relevant difference between study groups in the primary outcome, we have combined hospital readmission with death. Preliminary studies suggest that sufficient numbers of children will reach this combined outcome. However, we have incorporated an interim analysis by the DSMC to review the accrued data, and an adapted sample size could be considered if the combined event frequency is less than predicted. It is possible that since most children receive antibiotics during hospitalisation, the benefit anticipated with the use of azithromycin based on previous trials of mass drug administration will not be observed. However, most hospitalised children are treated with penicillins, cephalosporins, gentamicin or cotrimoxazole while in hospital, and the broad spectrum of activity (including malaria prevention) and long half-life of azithromycin suggest that there may be additive treatment and/or prophylactic benefit. Similarly, children may receive azithromycin during follow-up—either as treatment for an illness or because the caregiver sought out azithromycin on learning of the hypothesis—and this azithromycin use may lead to contamination in the placebo arm. After discharge, it is difficult to ensure adherence with the full 5-day treatment course. We will measure adherence using three different measures (text message responses, bottle check boxes and caregiver report at follow-up visits) although all are limited by caregiver report. However, the mortality benefit of azithromycin observed in Ethiopia was from a single dose[16] and in this study, the first dose will be directly observed. In addition, while relying on caregiver report of mortality and morbidity may lead to bias due to outcome misclassification, this misclassification should not differ between randomisation arms and therefore will be non-differential. Further hospital records will be used when available to determine diagnoses. Finally, resistance prevalence may be lower than predicted, limiting power to detect clinically relevant differences in resistance prevalence between the intervention arms. We will store all isolates

in the event that a greater number of isolates are needed for antibiotic resistance testing.

## Ethics and dissemination

The clinical trial is registered with clinicaltrials.gov (NCT02414399). Any modifications to the study protocol or consent materials will be submitted for approval by all regulatory authorities before implementation. The study is being externally monitored, and a DSMC has been assembled to monitor patient safety and to evaluate the efficacy of the intervention. Results of this study will be disseminated by publication in a peer-reviewed scientific journal, presented at relevant academic conferences and among participating partners and health facilities in Kenya.

**Author affiliations**
[1]Department of Global Health, University of Washington, Seattle, Washington, USA
[2]Centre for Clinical Research, Kenya Medical Research Institute, Nairobi, Kenya
[3]Childhood Acute Illness and Nutrition Network, Nairobi, Kenya
[4]Department of Epidemiology, University of Washington, Seattle, Washington, USA
[5]Department of Pediatrics, University of Washington, Seattle, Washington, USA
[6]Department of Allergy and Infectious Disease, University of Washington, Seattle, Washington, USA
[7]Department of Biostatistics, University of Washington, Seattle, Washington, USA
[8]Department of Pharmacy, University of Washington, Seattle, Washington, USA
[9]Centre for Microbiology Research, Kenya Medical Research Institute, Nairobi, Kenya
[10]Department of Pediatrics, School of Medicine, University of Nairobi, Nairobi, Kenya

**Acknowledgements** Pfizer donated the Zithromax to be used in this clinical trial, and Copan Diagnostics donated all rectal swabs and Cary-Blair media. Investigators from KEMRI-Wellcome Trust Kilifi, Jay Berkley, Anthony Scott, Joseph Waichungo, Angela Karani, Donald Akech and Horace Gumba provided microbiology expertise and training in nasopharyngeal swab collection, STGG media preparation and laboratory quality assurance and control. Liru Meshack Wekesa and George Bogonko provide clinical expertise and facilitate the study's integration into paediatric wards at two of the study hospitals, Homa Bay District and Kisii Teaching & Referral Hospital, respectively. Alex Awuor and Caleb Okonji, with the support of Richard Omore, provided training in anthropometric measurement. We are extremely thankful to Dr Philip Walson, who developed azithromycin dosing regimens. Hannah Atlas and Stephanie Belanger contributed to the standard operating procedure and case report form development and implementation. Jeremiah Moon created the visual instruction sheets for drug administration. Gillian Levine played an invaluable role in the proposal development.

**Contributors** JW, PBP, GJ-S, BAR, BOS and RN conceived of this trial and developed the study protocol. JW and BOS are study coprincipal investigators, and PBP is the project director; BAR oversaw the statistical analyses plans; JBB developed the CEA plan; KDT developed procedures for ascertaining and reporting SAEs; CJM developed procedures related to blood specimen procedures and drug adherence measurement. GJ-S, CJM, RN and PBP provided scientific expertise. RLB and MA are involved in collection and management of the data. MA and PBP coordinated and oversaw implementation of all clinical study procedures, and SK, with assistance from DR, oversees all laboratory procedures. All authors contributed to the development of this manuscript and/or study procedures and to reading and approving the final version for publication.

**Funding** This work was funded by the National Institutes of Health Eunice Kennedy Shriver National Institute of Child Health & Human Development, grant number R01 HD079695.

**Competing interests** None declared.

**Ethics approval** This study has received IRB approval by the University of Washington Human Subjects Division, KEMRI Scientific and Ethics Review Unit and the Kenya Pharmacy and Poisons Board.

**Provenance and peer review** Not commissioned; externally peer reviewed.

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
