## [Reviewer comments · BMJ Open]

ARTICLE DETAILS

TITLE (PROVISIONAL)	Azithromycin to prevent post-discharge morbidity and mortality in Kenyan children: A protocol for a randomized, double-blind, placebo-controlled trial (the Toto Bora trial)
AUTHORS	Pavlinac, Patricia; Singa, Benson; John-Stewart, G; Richardson, BA; Brander, Rebecca; McGrath, Christine; Tickell, Kirkby; Amondi, Mary; Rwigy, Doreen; Babigumira, Joseph; Kariuki, Sam; Nduati, Ruth; Walson, Judd

VERSION 1 – REVIEW

REVIEWER	Matthew Wiens University of British Columbia, Canada Mbarara University of Science and Technology, Uganda
REVIEW RETURNED	06-Sep-2017

GENERAL COMMENTS	Overall: This is an important and interesting study. It is clear that post-discharge mortality in the pediatric population is a neglected issue, and that novel interventions to improve outcomes are urgently required. I applaud Palvinac et al for this work and look forward to seeing the results soon. The overall justification and design are appropriate to answer the primary and secondary study objectives. There are, however, several minor issues which I have outlined below. Intro: Might be good to mention the study by Berkley et al on post-discharge mortality and malnutrition and use of Abx, to provide more context on the controversy of this approach. might be good to have some additional context on the importance of post-discharge mortality, various rates reported in different disease states, regions etc. Inclusion criteria: I wonder why the inclusion criteria does not specify that the child must have been admitted for an infectious illness? From the criteria outlined it appears that only injury/trauma and birth defects are exclusion criteria. What about cancer, asthma, sickle cell pain crisis, and other diseases that do not fall under the exclusion criteria?
--

Outcomes:

One concern I have for the combined outcome of death and/or re-admission is that re-admission is not as good of a proxy of recurrent illness as it would be in a more developed country settings. It has been shown that more than 50% of deaths following discharge do not occur within health facilities (i.e. mostly at home). Therefore, many children who become acutely ill, but do not die, are never admitted. This then can lead to re-admissions not being a good proxy of severe recurrent illness.

Is any data on recurrent illness not leading to admission being collected? Perhaps an additional secondary endpoint would be death/recurrent severe illness, where recurrent severe illness can be defined as re-admission or illness at home resulting in some pre-defined criteria that reasonably reflects severe illness as reported by caregivers.

Data collection:

Are you collecting any data on whether or not the child was re-exposed to azithromycin in the community, after discharge? I may suspect that children who become ill following admission may be exposed to azithromycin through the efforts of the caregiver in caring for the child. Since the study information (consent form, etc.) presumably describes that not all children are going to receive this drug, and also provides the name of the drug, and also may give the impression that this drug may improve health outcomes, some caregivers may seek this drug out to provide to their child (something that is not difficult in your context).

It might be good to see a table of the data which will be collected at discharge and during each of the 2 follow-up visits.

Analysis:

How will you consider known post-discharge exposure to azythromycin in the analysis?

When looking at carriage, how will you consider re-exposures to other antibiotic with potential activity against the pathogens of interest?

Statistical power:

The text under Figure 1 is hidden by the figure. However, this is standard language and I do not see any issues with how the power calculations were done.

Challenges and Limitations:

I think that a primary challenge/limitation of this study is the potential for contamination of the placebo group with antibiotics (including azithromycin) that is due to the very fact that this is an study of post-discharge use of antimicrobials. The hypothesis is thus known to caregivers and the access to azithromycin and other similar antibiotics is easy and inexpensive. This potential contamination affects all outcomes and should probably be mentioned somewhere. Since indiscriminate antibiotic use is already rampant in East Africa, this should be a concern to investigators.

	The MAL-ED cohort found that children are exposed to, on average, 5 courses of antibiotics per year (http://www.who.int/bulletin/volumes/95/1/16-176123/en/). For the outcomes of carriage/resistance, measured at fixed time points (3m and 6m), it is possible that a relationship between the interventions clinical effectiveness and these outcomes will be seen. If clinically effective, the control group is likely to have a higher rate of late exposure to antimicrobials (closer to time of evaluation) due to recurrent infectious illness, and if effective against the pathogens of interest, will dilute the potential effect that would have been seen if no clinical effect were to be observed.
--	---

REVIEWER	Eric Houpt University of Virginia
REVIEW RETURNED	13-Sep-2017

GENERAL COMMENTS	I reviewed the manuscript “Azithromycin to prevent post-discharge morbidity and mortality in Kenyan children: A protocol for a randomized, double-blind, placebo-controlled trial” It is clearly written and describes the methodology of the trial in excellent detail, which I believe is the primary purpose of BMJ Open (not for criticisms or suggestions for the trial, whose protocol is already approved if not underway). Some general comments, however: An interesting and practical trial, look forward to the results. Why not single dose azithromycin, for feasibility. One key reason stated for targeting hospitalized children up on discharge is that these children are high risk for death but are accessible to health care intervention (whereas treating the entire childhood population with azithromycin is less feasible). If this study shows a positive effect how generalizable will this be? This will work for regional hospitals in Kenya, but probably not Aga Khan hospital? Will it work for an intermediate economy such as South Africa. Will it work in these hospitals in Kenya for years to come, since health care indices may be improving. Where will the line be drawn and what would the recommendation be to MOH. The ancillary and subgroup analyses may be revealing, since it is likely that ALL hospitalized children is overly broad but there are subsets at risk of infectious mortality. And this could perhaps lead to a more generalizable recommendation. Since the AMR inducing effect of azithromycin has been seen in so many earlier studies, measuring this is fine but is also replicative. Yes it will show higher rates of resistance in S pneumo and maybe E coli, but the main question is whether this is deleterious, which would require a larger study presumably.
--

VERSION 1 – AUTHOR RESPONSE

REVIEWER #1

Reviewer comment: This is an important and interesting study. It is clear that post-discharge mortality in the pediatric population is a neglected issue, and that novel interventions to improve outcomes are urgently required. I applaud Pavlinac et al for this work and look forward to seeing the results soon. The overall justification and design are appropriate to answer the primary and secondary study objectives. There are, however, several minor issues which I have outlined below.

Author response: Thank you very much for your support of this trial.

Reviewer comment: Might be good to mention the study by Berkley et al on post-discharge mortality and malnutrition and use of Abx, to provide more context on the controversy of this approach.

Author response: Thank you for this suggestion. We agree that this important and relevant study should be mentioned. We have added a brief discussion of this research in the Background section of the manuscript.

Reviewer comment: might be good to have some additional context on the importance of post-discharge mortality, various rates reported in different disease states, regions etc.

Author response: Thank you for this suggestion. We have added a summary of the epidemiology of post-discharge mortality reported in prior literature from the Sub-Saharan African region in the Background section of the manuscript.

Reviewer comment: I wonder why the inclusion criteria does not specify that the child must have been admitted for an infectious illness? From the criteria outlined it appears that only injury/trauma and birth defects are exclusion criteria. What about cancer, asthma, sickle cell pain crisis, and other diseases that do not fall under the exclusion criteria?

Author response: Thank you for raising this important point. The mechanisms by which azithromycin reduces mortality are not well understood. For example, in the large trial of community mass drug administration of azithromycin conducted in Ethiopia, mortality benefit was observed despite a non-targeted approach. Azithromycin may be acting by treating undiagnosed or incompletely treated infection but may also reduce mortality through non-infection related mechanisms such as improved enteric function, prevention of future infection, reductions in inflammation, or decreases in immune activation. As a result, we do not exclude children with non-infectious conditions, as many children may benefit from the azithromycin intervention.

Reviewer comment: One concern I have for the combined outcome of death and/or re-admission is that re-admission is not as good of a proxy of recurrent illness as it would be in a more developed country settings. It has been shown that more than 50% of deaths following discharge do not occur within health facilities (i.e. mostly at home). Therefore, many children who become acutely ill, but do not die, are never admitted. This then can lead to re-admissions not being a good proxy of severe recurrent illness.

Author response: Thank you for this thoughtful comment. We agree this is a limitation of using re-hospitalization as a surrogate of illness but also feel as though the reviewer comment is an important component of the rationale for using re-hospitalization as a surrogate marker of death. The illnesses that do result in hospitalization likely represent the most severe cases of disease given the known barriers to reaching, and being admitted to, hospitals. Hospitalizations are also very costly to health care systems and individual families. Further, we don't expect the barriers to hospitalization to be differential across the treatment groups therefore any bias will result in a conservative estimation of the true treatment effect. This being said, we will be sure to raise this important limitation to the re-hospitalization outcome in the manuscript when the study concludes.

Reviewer comment: Is any data on recurrent illness not leading to admission being collected? Perhaps an additional secondary endpoint would be death/recurrent severe illness, where recurrent severe illness can be defined as re-admission or illness at home resulting in some pre-defined criteria that reasonably reflects severe illness as reported by caregivers.

Author response: Study staff encourage caregivers to bring the child back to the health facility if the child becomes ill, and these unscheduled visits, along with our scheduled follow up visits, present multiple opportunities for morbidity assessment (including adverse event monitoring) throughout the trial. Using standardized case report forms, we collect data on recurrent illnesses the child experiences during follow up (ascertained by caregiver recall for past illnesses, and by physical exam by the study clinician for current illnesses), as well as data on barriers to health seeking. These morbidity outcomes will be reported as secondary outcomes. Because of challenges with participant recall of illness timing and severity, and the intentional infrequency of scheduled follow-up visits to most mimic the real-world health-care seeking setting, we decided not to use a combined death/self-reported severity outcome. However, we have added, as a secondary endpoint, mild, moderate, and serious events (diarrhea, vomiting, fever, skin rash, difficulty breathing/wheeze, and seizure) that did not lead to a re-hospitalization and associated severity reported by caregivers at scheduled follow-up or identified by the study team during clinical exams.

Reviewer comment: Are you collecting any data on whether or not the child was re-exposed to azithromycin in the community, after discharge? I may suspect that children who become ill following admission may be exposed to azithromycin through the efforts of the caregiver in caring for the child. Since the study information (consent form, etc.) presumably describes that not all children are going to receive this drug, and provides the name of the drug, and also may give the impression that this drug may improve health outcomes, some caregivers may seek this drug out to provide to their child (something that is not difficult in your context).

Author response: This is a very interesting point. We are collecting antibiotic use data and we will include the frequency of caregiver-reported post-discharge antibiotic use, including azithromycin. The primary analysis of this study will be an intention-to-treat analysis, and therefore we cannot condition inclusion in this analysis on post-discharge azithromycin use. However, we have added to the protocol that we will conduct sensitivity analyses excluding children who report any antibiotic use, or specifically azithromycin use, over follow-up. However, we expect non-study azithromycin use to be fairly uncommon. The cost of azithromycin in Kenya is relatively high (ranging from 450 Kenyan Shillings (or approximately \$4.50 USD) to 1700 Kenyan Shillings (or approximately \$17 USD) estimated cost in Kenyan shillings for 5-day course), and data from the MAL-ED cohort in the East Africa Tanzania site suggests that macrolide use in this region is quite rare.¹

Reviewer comment: It might be good to see a table of the data which will be collected at discharge and during each of the 2 follow-up visits.

Author response: Thank you for this suggestion, and we agree that it will be helpful for readers to understand the data we collect at each visit. We have added a table summarizing the data collection at each study visit (now Table 2):

Reviewer comment: How will you consider known post-discharge exposure to azithromycin in the analysis?

Author response: Thank you for raising this important comment. As noted above, azithromycin appears to be infrequently used in the study area and we do not expect this to be a significant issue. However, we recognize that there could be contamination in our placebo group by azithromycin being used outside of the study protocol. We exclude children who are prescribed macrolide antibiotics at hospital discharge to avoid the likelihood of azithromycin use at discharge, but as the reviewer points out, we cannot control whether or not caregivers obtain the antibiotic at local pharmacies.

If this intervention were to be considered by policy-makers, the magnitude of benefit of the intervention would need to be large enough to outweigh the potential attenuating effect of antibiotic use in the placebo-assigned treatment arm. Further, we hypothesize that the timing of the azithromycin use, at discharge and the days immediately after, is a particularly important time for the antibiotic, as the child may be particularly susceptible to new and recrudescing infections. To explore this issue further, we have proposed sensitivity analyses in which we will exclude children whose caregivers report post-discharge antibiotic, and specifically azithromycin use. If we don't find an effect of the intervention and find that many children in the placebo-arm were using azithromycin, then we will report this as one plausible explanation for the null result.

Reviewer comment: When looking at carriage, how will you consider re-exposures to other antibiotics with potential activity against the pathogens of interest?

Author response: The reviewer has raised the important point that antibiotic resistance could be caused both by the intervention and any other antibiotic use during the follow-up period. If the post-discharge course of azithromycin does indeed prevent infectious illness, and as a consequence reduces antibiotic use, it is plausible that the prevalence of resistance to commonly used antibiotics will be higher (or equal) in the placebo group than in the treatment group as a result of other antibiotic use rather than an increase in resistance in the treatment group. Such an effect would be important to demonstrate as it would provide an accurate reflection of the real-world situation if the intervention were to be implemented and would highlight that reduction in illness plays an important role in antibiotic stewardship, even if it means increasing the indication for antibiotics in specific high-risk populations. Similarly, if azithromycin resistance is more common in the azithromycin group, despite frequent macrolide use in the placebo-treated group, then that would demonstrate the true risk to a 5-day course of azithromycin. We will be evaluating azithromycin-specific resistance, as well as resistance to other commonly used antibiotics. Because azithromycin use is rare in these settings, we don't expect azithromycin use to be common in the placebo group.

Reviewer comment: Statistical power: The text under Figure 1 is hidden by the figure. However, this is standard language and I do not see any issues with how the power calculations were done. Author response: We apologize for this formatting error, and have ensured this is fixed in the revision.

Reviewer comment: Challenges and Limitations: I think that a primary challenge/limitation of this study is the potential for contamination of the placebo group with antibiotics (including azithromycin) that is due to the very fact that this is an study of post-discharge use of antimicrobials. The hypothesis is thus known to caregivers and the access to azithromycin and other similar antibiotics is easy and inexpensive. This potential contamination affects all outcomes and should probably be mentioned somewhere. Since indiscriminate antibiotic use is already rampant in East Africa, this should be a concern to investigators. The MAL-ED cohort found that children are exposed to, on average, 5 courses of antibiotics per year (<http://www.who.int/bulletin/volumes/95/1/16-176123/en/>).

Author response: Thank you for raising this important point, and we agree that frequent antibiotic use is common among children in sub-Saharan Africa. In Kenya, azithromycin-specifically is rarely used (erythromycin is the most commonly used macrolide antibiotic). We agree that potential contamination is important to raise as a limitation, particularly as an explanation for no treatment effect if such a finding is made. However, if a benefit is observed, and that the benefit is sustained even when compared to a group of children treated with frequent antibiotics, will make for more compelling evidence to move towards implementing such a strategy.

Reviewer comment: For the outcomes of carriage/resistance, measured at fixed time points (3m and 6m), it is possible that a relationship between the interventions clinical effectiveness and these outcomes will be seen. If clinically effective, the control group is likely to have a higher rate of late exposure to antimicrobials (closer to time of evaluation) due to recurrent infectious illness, and if effective against the pathogens of interest, will dilute the potential effect that would have been seen if no clinical effect were to be observed.

Author response: We agree that there is a risk of antibiotic contamination in the placebo group and have tried to address these concerns in earlier responses.

Reviewer comment: I reviewed the manuscript "Azithromycin to prevent post-discharge morbidity and mortality in Kenyan children: A protocol for a randomized, double-blind, placebo-controlled trial". It is clearly written and describes the methodology of the trial in excellent detail, which I believe is the primary purpose of BMJ Open (not for criticisms or suggestions for the trial, whose protocol is already approved if not underway). Some general comments, however: An interesting and practical trial, look forward to the results.

Author response: Thank you for your review of this protocol.

REVIEWER #2

Reviewer comment: Why not single dose azithromycin, for feasibility.

Author response: Thank you for this important question. When designing the dosage, we determined that providing a therapeutic dose of 5-days would maximize the likelihood of seeing an effect if an effect truly exists. If a single dose did not show an effect, we would not be able to exclude the possibility that longer courses would be beneficial. We are also collecting adherence data from caregiver self-report (via SMS and questionnaire) and will conduct sensitivity analyses comparing the magnitude of effect in children reported to have received a full course vs. different numbers of doses.

Reviewer comment: One key reason stated for targeting hospitalized children on discharge is that these children are at high risk for death but are accessible to health care intervention (whereas treating the entire childhood population with azithromycin is less feasible). If this study shows a positive effect how generalizable will this be? This will work for regional hospitals in Kenya, but probably not Aga Khan hospital? Will it work for an intermediate economy such as South Africa. Will it work in these hospitals in Kenya for years to come, since health care indices may be improving. Where will the line be drawn and what would the recommendation be to MOH.

Author response: Thank you, the issue of sustainability and implementation of this intervention is an important point to raise. We intentionally implemented minimal exclusion criteria for the population in our study to be as generalizable as possible. We expect this intervention to be effective in any context in which post-discharge mortality and re-admissions are high; in which there are vulnerable populations such as malnourished or HIV-infected children; in which the availability of medical supplies is limited leading to infections that are undiagnosed and untreated; or in which nosocomial and community-acquired infections are prevalent. In the cost-effectiveness analyses, we will vary rates of antibiotic use, re-hospitalization rates, and mortality rates to establish thresholds of these variables that make this intervention no longer cost-effective. We do recognize the reviewer's important point that health care services and public health conditions may be improving over time, potentially rendering this intervention less effective or irrelevant as this happens. However, in the meantime, discharge azithromycin may save thousands of lives and understanding the mechanism by which it is working could shed important light into new interventions for post-discharge morbidity and mortality.

Reviewer comment: The ancillary and subgroup analyses may be revealing, since it is likely that ALL hospitalized children is overly broad but there are subsets at risk of infectious mortality. And this could perhaps lead to a more generalizable recommendation.

Author response: This is a very important point, and we thank the reviewer for raising it. We agree that these subgroup analyses may be informative in understanding the populations at risk for post-discharge infectious mortality, and identifying groups (based on clinical, pathogen, or host factors) in which the azithromycin intervention is more or less effective. This will also be useful for targeting the intervention if it were to be implemented. We agree with the reviewer on the value of these subgroup analyses, and will thoroughly conduct and present the results of these analyses in the final manuscript and to decision-makers.

Reviewer comment: Since the AMR inducing effect of azithromycin has been seen in so many earlier studies, measuring this is fine but is also replicative. Yes it will show higher rates of resistance in S pneumo and maybe E coli, but the main question is whether this is deleterious, which would require a larger study presumably.

Author response: Thank you for this comment. We actually are not convinced that we will see resistance increase in azithromycin-treated children at the two follow-up timepoints of 3 and 6 months. Most of the azithromycin resistance literature that we have found is from community-wide mass-drug administration distribution trials and as the reviewer notes, these trials do observe an increase in azithromycin resistance in S. pneumoniae² and E.coli³ in the 6 months after community-wide administration. The selective pressure from a mass-drug administration trial would be much stronger than the targeted approach that is being used in this trial, and thus the resistance development in individuals treated with azithromycin, when living in a community (and environment) without selective pressures from mass administration, we believe to be an unanswered question. If we do observe more resistance in S.pneumoniae or E.coli in the azithromycin-treated children or contacts of treated children, the clinical relevance of such resistance would need further study, as the reviewer notes. Further, we may observe more resistance in the placebo-treated children, due to less incurrent illness, which would be a novel finding with public health relevance. If the trial has no clinical benefit, then the question of resistance may be of little value as the strategy would not be implemented. However, if the trial does show clinical benefit, and resistance is more common in treated children or their contacts, then further longitudinal studies establishing the clinical relevance of resistance will be needed.

REFERENCES

1. Rogawski ET, Platts-Mills JA, Seidman JC, et al. Use of antibiotics in children younger than two years in eight countries: a prospective cohort study. *Bull World Health Organ* 2017;95(1):49-61. doi: 10.2471/blt.16.176123 [published Online First: 2017/01/06]
2. Coles CL, Mabula K, Seidman JC, et al. Mass distribution of azithromycin for trachoma control is associated with increased risk of azithromycin-resistant *Streptococcus pneumoniae* carriage in young children 6 months after treatment. *Clin Infect Dis* 2013;56(11):1519-26. doi: 10.1093/cid/cit137
3. Seidman JC, Coles CL, Silbergeld EK, et al. Increased carriage of macrolide-resistant fecal *E. coli* following mass distribution of azithromycin for trachoma control. *International journal of epidemiology* 2014;43(4):1105-13. doi: 10.1093/ije/dyu062 [published Online First: 2014/03/25]

VERSION 2 – REVIEW

REVIEWER	Matthew Wiens University of British Columbia, Canada
REVIEW RETURNED	10-Oct-2017
GENERAL COMMENTS	My concerns as outlined in my prior review have been sufficiently addressed. Thank you for allowing me to review this manuscript.

VERSION 2 – AUTHOR RESPONSE

Thank you very much for your continued review and editorial suggestions. In this revision, we have made the requested formatting changes and hope you find them satisfactory. Please don't hesitate to let us know if we can provide any further assistance. Thank you again and we look forward to your response.